# Management of Acute Bronchiolitis in Spoke Hospitals in Northern Italy: Analysis and Outcome

**DOI:** 10.3390/diseases12010025

**Published:** 2024-01-16

**Authors:** Carla Guidi, Neftj Ragusa, Ilaria Mussinatto, Francesca Parola, Diego Luotti, Giulia Calosso, Eleonora Rotondo, Virginia Deut, Fabio Timeus, Adalberto Brach del Prever, Massimo Berger

**Affiliations:** 1Pediatrics Department, Ciriè Hospital, 10073 Ciriè, TO, Italy; cguidi@aslto4.piemonte.it (C.G.); francesca.parola@aslto4.piemonte.it (F.P.); abrachdelprever@aslto4.piemonte.it (A.B.d.P.); 2Pediatrics Department, Ivrea Hospital, 10015 Ivrea, TO, Italy; nragusa@aslto4.piemonte.it (N.R.);; 3Pediatrics Department, Chivasso Hospital, 10034 Chivasso, TO, Italy; imussinatto@aslto4.piemonte.it (I.M.); ftimeus@aslto4.piemonte.it (F.T.)

**Keywords:** bronchiolitis, children, respiratory syncytial virus, high flow nasal cannula

## Abstract

Bronchiolitis is an acute viral infection of the lower respiratory tract that affects infants and young children. Respiratory syncytial virus (RSV) is the most common causative agent; however, other viruses can be involved in this disease. We retrospectively reviewed the clinical features of infants aged less than 12 months hospitalized for acute bronchiolitis in our Pediatric Units of Chivasso, Cirié, and Ivrea in Piedmont, Northern Italy, over two consecutive bronchiolitis seasons (September 2021–March 2022 and September 2022–March 2023). Patient-, disease-, and treatment-related variables were analyzed. The probability of therapeutic success (discharge home) was 96% for all patients (93% for RSV vs. 98% for non-RSV patients, *p* > 0.05). Among 192 patients, 42 infants (22%) underwent high-flow oxygen support (HFNC), and only 8 (4%) needed to be transferred to our hub referral hospital. Factors associated with hub hospital transfer were the age under 1 month and the failure of HFNC. The wide and increasing use of HFNC in pediatric inpatients improved the management of bronchiolitis in Spoke hospitals, reducing transfer to a hub hospital provided with Intensive Care Units.

## 1. Introduction

Bronchiolitis is an acute viral infection of the lower respiratory tract and is the leading cause of infant hospitalization in developed countries [1]. Although it can be diagnosed in adults and the elderly, children are the most affected by the disease, accounting for up to 15–17% of all hospitalizations in children younger than 2 years and 15% of emergency room admissions in infants [2]. Bronchiolitis is generally a self-limiting condition, but it can lead to severe respiratory distress and potentially culminate in acute respiratory failure, especially in infants aged less than 12 months, preterm infants (born at gestational age less than 37 weeks), or with underlying comorbidities, such as chronic lung or heart diseases, neurological conditions, immunodeficiency, exposure to cigarette smoke, and disadvantaged socioeconomic status [3,4,5].

Respiratory syncytial virus (RSV) is the most common causative agent, accounting for 60–80% of all cases. Rhinovirus (RV), Parainfluenza virus, Influenza virus, Metapneumovirus (MPV), and Adenovirus have also been reported [6,7,8,9]. RSV can also cause severe acute respiratory infections among high-risk groups such as pregnant women, the elderly, and individuals with underlying medical conditions [10].

Seasonal variations in RSV infectivity, which is higher during winter months in temperate zones, have been well described [11,12,13].

Breastfeeding is associated with lower incidence and severity of lower respiratory tract disease in infants, and several studies have confirmed how a longer duration of breastfeeding is associated with better clinical outcomes of bronchiolitis [14]. Viral bronchiolitis severity is linked to innate immunity. In RSV-infected infants, breastfeeding enhances immune response by reducing neutrophilic airway infiltration and immune modulators (airway chemokines and IL8) and increasing interferon-α [15].

Diagnosis of bronchiolitis is based on clinical history and physical examination [3]. Since there is no effective treatment for bronchiolitis, cornerstones of management are mostly supportive, consisting of fluid management and respiratory support [3,4,5,14,16]. Pharmacological interventions, such as nebulized bronchodilators, local or systemic steroids, and antibiotics, have limited or no evidence of efficacy and generally are not recommended.

Standard oxygen supplementation in bronchiolitis is provided by nasal cannula (NC) and is upgraded to continuous positive airway pressure (CPAP) or invasive ventilation if needed [17]. In recent years, High-Flow Nasal Cannula oxygen therapy (HFNC), a non-invasive respiratory support supply, has emerged as a promising method for providing oxygen to children with bronchiolitis [18]. Initially restricted to neonatal and pediatric intensive care units (NICUs and PICUs), over the last decade, HFNC has migrated to emergency departments, inpatient pediatric wards, and patient transfers [19,20,21].

Evidence suggests that the use of HFNC in bronchiolitis is limited to rescue therapy after failure of standard NC only in infants who are hypoxic [22]. Different from previous, recent available data show no significant benefits for children treated with HFNC compared with NC or CPAP and suggest that HFNC produces no relevant differences in duration of hospitalization, days with oxygen supplementation, and rate of admissions to PICUs compared to NC [19,21,23,24]. Furthermore, some observational studies suggest that HFNC reduces respiratory work and may decrease the need for intubation but does not reduce the intensive care rate of admissions [25,26].

However, more studies are needed to evaluate the role of HFNC outside of the PICU setting, as there is still a lack of guidelines to standardize the use of HFNC in children with respiratory failure.

As treatments are limited, preventive measures are needed to reduce morbidity and mortality, especially from RSV infections. Environmental measures are crucial in preventing and limiting the spread of bronchiolitis (i.e., performing frequent hand washing and surface cleaning, avoiding tobacco exposure or crowded places, and limiting contact with subjects suffering from respiratory infection symptoms). Moreover, pharmacological immunoprophylaxis with palivizumab has been proven beneficial to populations at increased risk for RSV infection-related complications, such as preterm babies and those affected by chronic lung or heart diseases, and it is currently available only for these specific infants [10]. New preventive approaches under development include maternal vaccines, other monoclonal antibodies (i.e., Nirsevimab, MK-1654), and pediatric vaccines. Maternal vaccines and monoclonal antibodies (mAb) protect infants during their first months of life through the transplacental transfer of maternal antibodies or by providing immediate protection for neonates. Pediatric vaccines represent a promising strategy for longer-lasting protection in infants >6 months of age (not in neonates and young infants entering their first RSV season) [10].

Recently, nirsevimab, a new mAb for the prevention of RSV infection, has been approved for use in Europe in infants under 1 year of age [27]. This mAb binds the RSV fusion protein, blocking viral entry into the host cell; differently from palivizumab, it has an extended half-life (5 months) covering the entire RSV season, and its use is not restricted to preterm infants. The Spanish Society of Paediatric Infectious Disease recently recommended routine administration of nirsevimab to all infants under 6 months of age who were born during the RSV season or under 6 months of age at the start of the winter season [28].

Our local health authority (ASL TO4) in Piedmont, North-West Italy, includes 174 municipalities, extending over a wide area of around 3165 square kilometers characterized by a great geographical variability, from high Alpine mountains to urban areas. The total population of this area is 504,467 people, and the healthcare system is organized into five districts, with great heterogeneity in demography, population density, geography, and infrastructure. The General Emergency Department and Pediatric Unit are present in three Spoke hospitals of Ciriè, Chivasso, and Ivrea. To provide uniformity of care with an evidence-based approach, in 2013, the first “Diagnostic Therapeutic Assistance Pathway for bronchiolitis in pediatric age” was drawn up. Afterward, it was updated in 2016 and 2020 in accordance with hub hospital and based on the national guidelines on bronchiolitis.

Every year a palivizumab administration program is carried out in our three Pediatric Units, according to the Italian Society of Neonatology recommendations.

All three Pediatric Units of ASL TO4 are equipped with high-flow oxygen supply systems.

The aims of the present study were (1) to describe the percentage of therapeutic success in bronchiolitis treated in Spoke hospitals of ASL TO4 over two consecutive bronchiolitis seasons, from 2021 to 2023, and (2) to evaluate clinical and biological differences between RSV positive (RSV+) and RSV negative (RSV−) patients, represented by patient-related factors (i.e., patient’s age, prematurity, and weight for gestational age) and infection-related factors (i.e., presence of fever, duration of supplemental oxygen therapy, both low-flow oxygen and HFNC, use of antibiotic or steroid therapy, intravenous hydration, and the need to be transferred to a hub hospital with a PICU or a NICU).

## 2. Materials and Methods

### 2.1. Study Design; Patient Enrollment

This study was conducted in three Pediatric Units of ASL TO4 Spoke hospital (Chivasso, Cirié, and Ivrea) in Piedmont, Northern Italy.

We retrospectively reviewed the clinical features of infants under 12 months of age referred to our Pediatric Units of Chivasso, Cirié, and Ivrea for acute bronchiolitis requiring hospitalization over two consecutive bronchiolitis seasons (from 1 September 2021 to 31 March 2022 and from 1 September 2022 to 31 March 2023).

We collected data about clinical and epidemiological characteristics (age at onset, gender, gestational age, birth weight, comorbidities, weight at admission, diet, previous respiratory infections, and fever), laboratory, microbiology and radiograph results, and short-term outcomes (length of hospital stay, complications, and the hub hospital transfer).

For etiological diagnosis, nasal swabs for RSV and nasopharyngeal swabs for SARS-CoV2 were performed in all patients. When available, nasopharyngeal swabs for Influenza virus, Adenovirus, and Rhinovirus were performed.

Blood cell count, C-reactive protein, and chest X-ray were performed according to clinical choice.

Treatment during hospitalization (low- or high-flow oxygen supplementation, nebulized therapy, steroids, antibiotics, and intravenous hydration) and discharge therapy data were recorded in all patients. All parents gave their consent for treatment and data registration in an anonymous dataset by signing a specific consent form.

This study was conducted in full conformity with the principles of the Declaration of Helsinki. In accordance with current regulations, this research is not among those requiring a formal opinion from the ethics committee. This is a secondary use of data for research purposes for which specific informed consent was requested ab initio from patients who would be undertaking a treatment process.

### 2.2. Statistical Analysis

Statistical analysis included only patients with the first episode of bronchiolitis; for patients hospitalized with a second episode of bronchiolitis, an extra summary table was added.

Univariate analysis was performed with the Chi-square or Fisher’s test for dichotomous variables; for continuous variables, the Kruskal–Wallis test for nonparametric measures was used. To define the probability of success, the Kaplan–Meier statistics estimator was used [29]. The difference between groups was calculated by log-rank test [30]. To identify independent factors (if any), we adopted logistic regression [31].

The univariate analysis was conducted by VassarStats (Statistical Computation Web Site), and the Kaplan–Meier statistics, together with the logistic regression by the NCSS software for Windows version 12 (https://www.ncss.com/ accessed on 5 September 2023), were used. A *p*-value below 0.05 was defined as statistically significant.

## 3. Results

Overall, 1331 patients were admitted to the three ASL TO4 Pediatric wards from 1 September 2021 to 31 March 2022 and from 1 September 2022 to 31 March 2023. Among these, 192 infants aged 0–12 months were admitted due to the first episode of acute bronchiolitis (14.4% of all hospitalizations, 43.6% of all hospitalizations in infants <12 months). The general characteristics of our study population are summarized in Table 1.

The median age of our patients was 2 months (range from 0 to 11 months), and males were more represented than females (60% vs. 40%). Gestational age was known in 90% of hospitalized children. Among these, preterm infants accounted for about 13%. Seventeen preterm patients (68%) had a gestational age between 35 + 0 and 36 + 6 weeks, while 7 (28%) had a gestational age between 30 + 0 and 34 + 6 weeks, and one (4%) was born at a gestational age less than 29 weeks. Bronchopulmonary dysplasia was detected in one patient (0.5% of all patients), while neither chronic heart, neurological conditions, nor immunodeficiency were recorded. A total of 136 infants (71% of all) were born appropriate for gestational age (AGA); 16 infants (8%) were small for gestational age (SGA), and 14 patients (7%) were large for gestational age (LGA). Breastfeeding was recorded in 123 infants (64%), while formula feeding was recorded in 43 patients (22%), and 17 infants (9%) had already been weaned when admitted to the hospital (data not available in 5%). No data about other risk factors for bronchiolitis (presence of siblings, smoking exposure, family’s socioeconomic status, etc.) were available in medical records.

Twelve patients (6% of all) had comorbidities (congenital kidney anomalies) or personal medical history of neonatal disorders (i.e., anemia, hypotonia, cerebral ventricular dilation, cephalocele, neonatal sepsis, neonatal invasive, or non-invasive respiratory support).

A pathogen was detected in 129 of the 192 patients (67%). The most detected agent was RSV (66% of all infants), without differences between the two study periods (65% in 2021–2022 and 67% in 2022–2023). Other observed pathogens were Influenza virus in one patient, Streptococcus pneumoniae in one patient, and Adenovirus, Rhinovirus, and Haemophilus influenzae simultaneously detected in another patient. One patient aged <1 month resulted positive for both SARS-CoV2 and RSV; one patient was positive for both RSV and Epstein–Barr virus.

One-fourth of all children presented with fever, with no differences between RSV-positive and negative patients (24% vs. 25%, respectively). Nebulized hypertonic saline (3%) was used in 77% of patients, and corticosteroids (oral or IV) were used in 13% of infants; no differences were observed between RSV-positive and negative patients. 

Laboratory, X-ray findings, and treatment during hospitalization are resumed in Table 2. Short-term hospitalization outcomes are shown in Table 3.

In our series, we could not find statistically significant differences between RSV-positive and RSV-negative groups, except for the greater use of antibiotics in the RSV-negative group (*p* = 0.004) and the higher probability of complications in the RSV-negative group (*p* = 0.03).

We recorded a trend for a higher need for HFNC in the RSV-positive patients (*p* = 0.06). In our cohort, we observed that 68% of RSV-positive patients (86/127) were breastfed. 

Figure 1 and Figure 2 report the probability of treatment success (discharge home without transfer to a hub hospital with PICU or NICU): 96% for all patients (95% CI 93–99); 94% (95% CI 87–98) for RSV-positive; and 98% (95% CI 95–100, *p* = NS) for RSV-negative patients when we stratify according to RSV infection.

A specific table describing the characteristics of infants with a second episode of bronchiolitis in the same timeframe was added (Table 4).

As reported in Table 4, nine infants had a second episode characterized by respiratory distress with crackles and wheezing over the observed period. Most of them were born at term, aged 1–3 months, breastfed, and resulted in RSV-negative. For these patients, the first bronchiolitis episode was RSV-positive in four patients (45%), while Rhinovirus was identified in one (11%).

Two patients had a personal medical history of neonatal disorders (cerebral ventricular dilation, neonatal sepsis). Three infants were febrile, and two underwent chest radiography. Nebulized hypertonic saline 3% was used in 67% of patients.

Due to the small number of patients, we did not observe statistically significant differences between RSV-positive and RSV-negative infants, except for HFNC, which was utilized more in RSV-positive patients than in RSV-negative ones.

Table 5 reports the univariate (Kaplan–Meier) analysis with the probability of treatment success of bronchiolitis (discharge home without transfer to a hub hospital with a PICU or a NICU) as the endpoint.

All infants who did not receive oxygen therapy or required only low-flow oxygen support were discharged home. Among patients treated with HFNC oxygen support, 19% needed to be transferred to a hub hospital (*p* = 0.005).

Patient’s age was the second statistically significant variable related to the outcome: patients under 1 month of age had a lower probability of treatment success (83%) than older infants (97% in children aged 1–3 months, 100% in children aged 4–6 months, and 94% in children older than 6 months; *p* = 0.045).

We observed a trend for a better outcome with the use of antibiotics (*p* = 0.06), which was more frequent in RSV-negative patients. The outcome was not related to the causative agent, gender, children’s growth, presence of fever, pCO_2_ value, or use of steroids or intravenous hydration.

On multivariable analysis, the patient’s age (*p* = 0.022) and the need for HFNC support (*p* < 0.0000001) have been confirmed to be independent factors (Table 6).

## 4. Discussion

Our study described the characteristics of infants under 12 months of age hospitalized for bronchiolitis in the three ASL TO4 Pediatric Units over two consecutive seasons (from 1 September 2021 to 31 March 2022 and from 1 September 2022 to 31 March 2023).

Most infants with bronchiolitis were previously healthy. Among risk factors, bronchodysplasia was detected in 0.5% of infants, and prematurity was detected in 13%. This rate was lower than those reported in recent Italian and USA studies (24.5% and 33%, respectively) [11,32]. Unlike these studies, our data were collected after the introduction of prophylaxis with palivizumab, which in Piedmont is administered to all infants at increased risk for RSV infection-related complications according to the recommendations of the Italian Society of Neonatology [4]. We might speculate that the lower percentage of preterm infants observed in our cohort might be the result of palivizumab prophylaxis that protects preterm infants from severe RSV infections. We might assume that this preventive measure was effective in our setting, given the 10% reduction. In Piedmont, based on recent epidemiological data, which showed an earlier peak of RSV infection, a new timing of the RSV prevention strategy has been proposed starting from 2022, providing for an early first dose of palivizumab in high-risk patients in October (rather than November).

Another factor that may have influenced the lower preterm rate in our study is the short distance between our Pediatrics Units and the hub hospital: infants might have direct access to the hub center where they underwent follow-up for prematurity or other risk conditions.

In this retrospective study, from available data in medical records, we were not able to collect information about other risk factors for bronchiolitis (siblings, smoking exposure, family’s socioeconomic status, etc.) that could influence the course of the disease.

We confirm the prevalent role of RSV as a causative agent of bronchiolitis requiring hospitalization (detected in 66% of hospitalized infants). Although severe prematurity is an important risk factor for severe bronchiolitis, in our study, children born late preterm and at term accounted, in absolute terms, for the largest proportion of hospitalizations due to RSV. This highlights the importance of protecting all neonates and children up to 1 year of age throughout the RSV epidemic season.

In our study, RSV-positive patients underwent less frequent administration of antibiotic therapy than RSV-negative ones: even if no differences were observed in patients with fever between the two groups, the different use of antibiotics was due to a more frequent finding of complications in the RSV-negative population such as pneumonia, bacterial superinfection, postinfectious neutropenia, and apnea. Among protective factors for infections, we observed that about half of RSV-negative patients were breastfed.

Breastfeeding plays a crucial role in protecting infants from infection by enhancing the immune response; a reduction in neutrophilic airway infiltration and immune chemokines and an increase in interferon-α were described in RSV-infected infants [15]. Recent data about breastfeeding in ASL TO4 collected during pediatric health checks in 2021 show that 68% of infants are exclusively breastfed in the first month of life, 60% at 2–3 months and 54% at 4–5 months. In our cohort, 64% of patients were breastfed; even if 67% of RSV-positive are exclusively breastfed, the absolute protective role of breastfeeding should be evaluated by analyzing the type of feeding among infants with bronchiolitis treated at home. Encouraging exclusive breastfeeding for at least six months is a pediatrician’s priority to reduce the morbidity of respiratory infections.

Despite the low prevalence of risk factors, more than 70% of children required oxygen supplementation, mostly low-flow oxygen. This is due to the fact that mild bronchiolitis is usually treated by Pediatricians at home or, eventually, in the hospital in Brief Intensive Observation areas before being discharged home.

A trend toward higher use of HFNC was observed more frequently in RSV-positive infants, presumably because they had a more severe clinical picture. An increased use of HFNC for bronchiolitis after the SARS-CoV-2 pandemic was observed in a recent Italian study [33]. In the pandemic period, these authors noticed an increase in RSV bronchiolitis (from 61% in 2018–2019 to 80% in 2021–2022) without a more severe clinical course (intubation rate and length of stay did not change), concluding that the increasing use of HFNC was mostly caused by a more aggressive therapeutic approach of pediatricians.

The restrictive measures adopted to contain the SARS-CoV2 pandemic (i.e., social distancing, use of face masks, and frequent hand washing) have changed the epidemiology of RSV infection [34]. In particular, a dramatic reduction in bronchiolitis hospitalizations was observed during the 2020–2021 season [35], and a subsequent unusual resurgence of RSV infection was described [36,37,38]. The prevalence of RSV subtype A was described in children in a study performed in Lombardy in children <2 years between 2009 and 2014, with no differences in bronchiolitis severity [39]. In a Sicilian study conducted between 2015 and 2020, children under 5 years of age and young adults (19–34 years) were more infected by the RSV-A subtype, while the RSV-B subtype was prevalent in adults (>35 years), in particular in hospital setting, among cases with underlying diseases and among those who developed a respiratory complication [40].

A recent Italian study of Istituto Superiore di Sanità demonstrated a change in RSV subtype circulation in the post-pandemic period [41]. In 2021–2022, the intense RSV peak was driven by RSV-A and was not associated with increased clinical severity of bronchiolitis in children younger than one year with respect to previous seasons. RSV-B predominance was instead reported in 2022–2023, causing more severe disease due, in part, to the RSV-B-specific immune debt and, in part, to the new RSV-B genetic strains that might have enhanced the RSV-B pathogenicity.

However, in our series, we did not observe an increase in RSV bronchiolitis during 2022–2023 in comparison with the previous one (2021–2022). In our region, it was not possible to obtain the RSV subtype.

Regarding the RSV epidemiological data, to date, there is no Italian national system for surveillance of hospitalizations suspected to be due to RSV. Starting with the 2019–2020 season, Italian national community-based RSV surveillance has been conducted through InfluNet (the Italian Influenza Surveillance Network), which has been active for several years and uses the influenza-like illness (ILI) case definition [42].

All transferred infants were refractory to HFNC, and the majority of them were RSV-positive (88% of cases). In our study, patients with a more severe clinical picture who required the use of HNFC had a similar duration of oxygen supply and hospitalization to that of patients with a milder clinical picture, confirming the usefulness of this therapeutic approach [24].

Despite a trend toward a higher incidence of RSV+ bronchiolitis in the total breastfed population, most RSV-positive infants less than <1 month who were transferred to hub hospital were fed infant formula. This is consistent with other studies in which human milk showed a protective role in RSV-positive infants. Jang et al. demonstrated a lower severity of RSV infection requiring oxygen therapy in breastfed hospitalized infants than in formula-fed ones [14]. We did not demonstrate a significant correlation between HFNC treatment and breastfeeding (*p* = 0.19).

Multivariable analysis showed that age <1 month and the need for HFNC support were independently associated with a worse clinical outcome, according to the literature [43,44]. In our setting, after the failure of low-flow oxygen administration, HFNC treatment represented a rescue therapy in bronchiolitis, permitting the treatment of moderate to severe bronchiolitis also in Pediatric Units in Spoke hospitals.

RSV bronchiolitis was a more severe illness, mostly in infants younger than 1 month of age. In our cohort of hospitalized infants with acute bronchiolitis, disease outcome was influenced by the infant’s age and type of treatment, the HFNC treatment being an indicator of poor outcome (need to transfer to a hub hospital). Infant’s age >1 month and no need for HFNC predicted a better outcome.

This study has some limitations. It is based on retrospective data from a limited sample of patients. The total number of hospitalized patients in ASL TO4 Spoke hospital may have been influenced by the proximity to the hub hospital to which some patients may have gone directly.

In addition, our retrospective study suffers from bias related to the lack of collected data and refers to hospitalized patients in a restricted geographical area, with its specific characteristics and seasonality, so results cannot be generalized. Although our results refer to a limited number of patients and a specific geographic area, they represent a starting point for targeted preventive actions, as our health district had a population of 3015 people under one year of age in 2022 [45]. Analysis of future seasons, also including bronchiolitis visits to the Emergency Department and discharged home and outpatients treated by general practitioners, will allow for a better understanding of the patterns of RSV infection in our area and to confirm our data.

## 5. Conclusions

The recent wide and increasing use of HFNC in pediatric inpatients improved the management of bronchiolitis in Spoke hospitals. Further studies are needed to analyze the impact of HFNC in reducing transfers from Spoke hospitals to hub hospitals equipped with intensive care units.

The observed changes in the epidemiology of RSV suggest the necessity of potentiating the epidemiological surveillance systems on RSV circulation. Information about seasonality, age, the severity of RSV infection, and at-risk populations are requested to face new epidemiological patterns and to program appropriate prevention strategies (i.e., timing of palivizumab prophylaxis).

Furthermore, also in our study, most hospitalizations for RSV occur in otherwise healthy infants born at term in the first months of age. New prophylactic strategies, including new mAb (i.e., Nirsevimab) and maternal and pediatric vaccines, are necessary to ensure that all neonates and infants are protected throughout the RSV season.

## Figures and Tables

**Figure 1 diseases-12-00025-f001:**
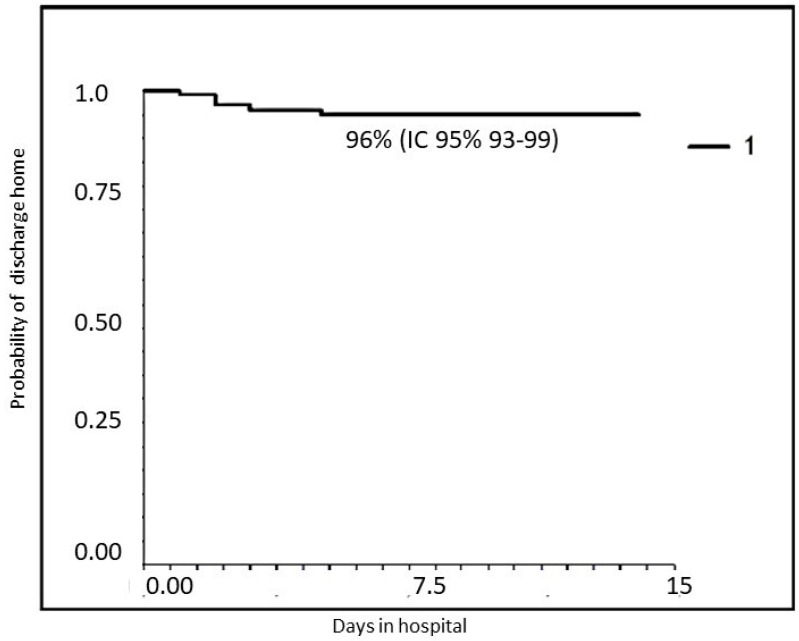
Probability of treatment success in the entire population (discharge to home).

**Figure 2 diseases-12-00025-f002:**
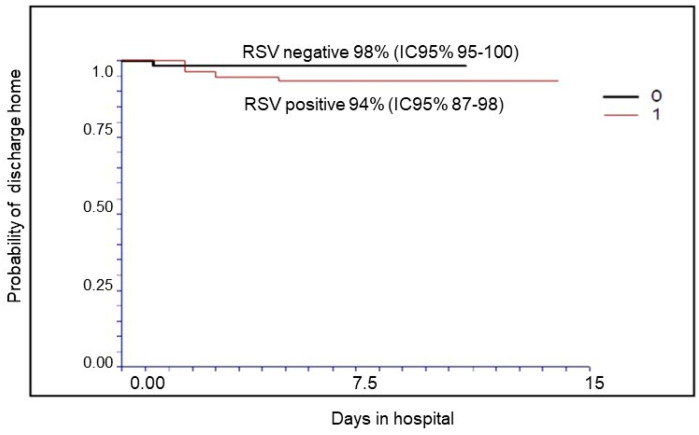
Probability of treatment success (discharge to home) in RSV-positive and RSV-negative patients.

**Table 1 diseases-12-00025-t001:** Demographic and epidemiological characteristics of infants with first episode of bronchiolitis.

		All Patients N (%) 192 (100)	RSV+ N (%) 127 (100)	RSV− N (%) 65 (100)	*p*-Value
Gender	Male	116 (60)	78 (61)	38 (58)	0.75
	Female	76 (39)	49 (39)	27 (42)	
Prematurity (<37 w)	Yes	25 (13)	14 (11)	11 (17)	0.28
	No	148 (77)	98 (77)	50 (77)	
	Unknown	19 (10)	15 (12)	4 (6)	
Age (months)	<1	27 (14)	19 (15)	8 (12)	0.32
	1-3	119 (62)	73 (57)	46 (71)	
	4-6	29 (15)	22 (17)	7 (11)	
	7-12	17 (9)	13 (10)	4 (6)	
Weight-for-age	AGA	136 (71)	87 (68)	49 (75)	0.13
	SGA	16 (8)	9 (7)	7 (11)	
	LGA	14 (7)	13 (10)	1 (1)	
	Unknown	26 (13)	18 (14)	8 (12)	

AGA: appropriate for gestational age. SGA: small for gestational age. LGA: large for gestational age. Prematurity: <37 weeks of gestational age.

**Table 2 diseases-12-00025-t002:** Laboratory, X-ray findings, and treatment during hospitalization.

		All Patients N (%)	RSV+ N (%)	RSV− N (%)	*p*-Value
Total (%)		192 (100)	127 (100)	65 (100)	
CRP higher level (mg/dL)			1.01 (0.03–60.3)	1.56 (0.03–78)	0.50
pCO_2_ higher level (mmHg)			47.5 (24.4–74)	47.3 (28–76.8)	0.76
Fever	Yes	46 (24)	30 (24)	16 (25)	1
	No	146 (76)	97 (76)	49 (75)	
Chest X-Rays	Yes	29 (15)	19 (15)	10 (15)	1
	No	163 (85)	108 (85)	55 (85)	
Low-Flow Oxygen	Yes	120 (62)	78 (61)	42 (65)	0.75
	No	72 (37)	49 (38)	23 (35)	
High-Flow Oxygen	Yes	42 (22)	33 (26)	9 (14)	0.06
	No	150 (78)	94 (74)	56 (86)	
High-Flow Oxygen days			6 (2–11)	6 (1–9)	0.3
Fraction of inspired Oxygen %			30% (25–65)	35% (25–60)	0.18
Antibiotics (oral, IV	Yes	4 (24)	23 (18)	24 (37)	0.004
	No	145 (75)	104 (82)	41 (63)	
Given hydration (IV)	Yes	31 (16)	22 (17)	9 (14)	0.67
	No	161 (84)	105 (83)	56 (86)	

CRP: C reactive protein. Continuous variables are expressed as medians and ranges. IV: intravenous hydratation.

**Table 3 diseases-12-00025-t003:** Short-term outcome hospitalization.

		All Patients N (%)	RSV+ N (%)	RSV− N (%)	*p*-Value
Total (%)		192 (100)	127 (100)	65 (100)	
Complications	Yes	24 (12)	11 (9)	13 (20)	0.03
	No	168 (88)	116 (91)	52 (80)	
Hospitalization days (median and range)			5 (2–14)	5 (1–11)	0.34
Transfer to a hub hospital	Yes	8 (4)	7 (5)	1 (2)	0.26
	No	184 (96)	120 (94)	64 (98)	

Complications were defined as events outside the normal course of the disease (i.e., peripheral venous access infection, seizures, appearance of vomiting, and diarrhea).

**Table 4 diseases-12-00025-t004:** Demographic and epidemiological characteristics of infants with a second episode of bronchiolitis.

		All Patients N (%) 9 (100)	RSV+ N (%) 3 (100)	RSV− N (%) 6 (100)	*p*-Value
Gender	Male	6 (67)	1 (33)	5 (83)	0.46
	Female	3 (33)	2 (67)	1 (17)	
Prematurity (<37 w)	Yes	1 (11)	1 (33)	0	0.33
	No	7 (78)	2 (67)	5 (83)	
	Unknown	1 (11)	0 (0)	1 (17)	
Age (months)	<1	0	0	0	0.31
	1-3	8 (89)	3 (100)	5 (83)	
	4–6	1 (11)	0 (0)	1 (17)	
	7–12	0 (0)	0 (0)	0 (0)	
CRP higher level (mg/dL)		3.3 (0.06–67.6)	1.9 (0.6–30)	12.3 (0.34–67.6)	0.33
pCO_2_ higher level (mmHg)		48.6 (41.1–61)	48.9 (41.1–59.9)	48 (42.2–61)	0.80
Hospitalization day		5 (2–11)	9 (2–11)	5 (3–6)	0.32
Low-Flow Oxygen	Yes	5 (55)	2 (67)	3 (50)	1
	No	4 (44)	1 (33)	3 (50)	
High-Flow Oxygen	Yes	3 (33)	3 (33)	0 (0)	0.011
	No	6 (67)	0 (0)	6 (100)	
High-Flow Oxygen days		5 (1–8)	5 (1–8)	NA	NA
Fraction of inspired Oxygen %		40 (25–50)	40 (25–50)	NA	NA
Need to transfer to a hub hospital	Yes	1 (11)	1 (33)	0 (0)	0.33
	No	8 (89)	2 (67)	6 (100)	

CRP: C reactive protein; it was expressed as medians and ranges, together with pCO_2_ levels, duration of HFNC treatment, fraction of O_2_, and airflow.

**Table 5 diseases-12-00025-t005:** Univariate analysis of probability of treatment success.

		Probability	95% CI	*p*-Value
All patients		96%	93–99	
Virus	RSV+	94%	87–98	0.26
	RSV−	98%	95–100	
Gender	Male	95%	91–99	0.40
	Female	96%	91–100	
Fever	Yes	98%	94–100	0.42
	No	95%	91–99	
pCO_2_ >50 mmHg	Yes	91%	88–93	0.03
	No	98%	95–100	
Low-Flow Oxygen	Yes	96%	93–100	0.25
	No	94%	89–100	
High-Flow Oxygen	Yes	81%	69–93	0.005
	No	100%		
Weight-for-age	AGA	95%	(91–99)	0.10
	SGA	100%		
	LGA	100%		
Diet	Breast milk	95%	92–99	0.85
	Infant formula	95%	89–100	
	Weaned baby	94%	83–100	
Antibiotics	Yes	100%		0.06
	No	94%	90–98	
Steroids	Yes	92%	81–100	0.59
	No	96%	93–99	
Patient’s age	<1 month	83%	69–99	0.045
	1–3 months	97%	95–100	
	4–6 months	100%		
	>6 months	94%	83–100	

AGA: appropriate for gestational age. SGA: small for gestational age. LGA: large for gestational age.

**Table 6 diseases-12-00025-t006:** Regression analysis of variables affecting the probability of failing therapies.

	SE	R2 Corrected	F	*p*-Value
Stratum age below 1 month	0.19	0.022	5.39	0.022
Presence of fever	0.2	0.003	0.59	0.44
RSV+	0.19	0.0088	1.69	0.14
Low-flow oxygen support	0.20	0.0028	0.55	0.45
High-flow oxygen support	0.18	0.155	34.92	<0.0000001
pCO_2_ above 50 mmHg	0.51	0.49	0.34	0.09

SE: standard error. F: risk coefficient.

## Data Availability

The collected data are available from corresponding authors.

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
