# Peer review of "Management of Acute Bronchiolitis in Spoke Hospitals in Northern Italy: Analysis and Outcome"

_diseases, 2024, doi:10.3390/diseases12010025_

Round 1
Reviewer 1 Report
Comments and Suggestions for Authors
Thank you for the invitation to revise the manuscript entitled "MANAGEMENT OF ACUTE BRONCHIOLITIS IN SPOKE HOSPITALS: ANALYSIS AND OUTCOME", an intriguing examination of the cases of bronchiolitis in three spoke centers of the ASL TO4, north-west of Italy, with some hints of analysis.
General comments
The manuscript definitely needs to be improved, and some details need to be expanded on.
Starting with the title, it is important to note that this is an evaluation conducted in three hospitals in northern Italy: it is always important to contextualize the study geographically. It should also be reformatted to remove all caps.
There are a few things that should be investigated. Among these are the importance of virological and epidemiological surveillance to better weigh the results of the many studies that, in the absence of a standard, do not provide awareness of the severity of the disease.
Specific comments
Abstract
Standardization of the text is required. It should also be noted that the study was conducted in Italy. The English is a little messy, and some points are unclear. Finally, (P=NS) is unclear; it should be written more clearly.
Keywords
A ";" should be used to separate each keyword. A running title appears as well, which is most likely a leftover from previous submissions.
1. Introduction
First and foremost, each paragraph should be reformatted in accordance with the journal's standards (by the way, the journal is "Diseases" and not "Vaccines"). Because the line numbers are missing, indicating the references will be more difficult.
- “Bronchiolitis is an acute viral infection of the lower respiratory tract that affects infants and young children.” à A reference must be inserted to support this sentence which, moreover, is inaccurate. Bronchiolitis is the inflammation of the bronchioles; that it is more frequently infectious and, in this case, viral is true, but let's not confuse the reader. It should also be noted that it primarily affects younger populations, but not exclusively.
- “It represents the most common cause of acute respiratory failure in infants (age <12 months) and the leading cause of infant hospitalization in developed countries.” à This sentence is repeated, in a very similar way, in the third paragraph “Bronchiolitis is generally a self-limiting condition, but can lead to severe respiratory distress and potentially culminate in acute respiratory failure, especially in infants aged less than 12 months […]” à Summarize and organize this section better.
- “Seasonal variations in RSV infectivity, that is the highest […]” à “higher” instead of “the highest”.
- “On the contrary, breastfeeding is associated with lower incidence and severity of lower respiratory tract disease and several studies have confirmed how longer duration of breastfeeding is associated with better clinical outcome of bronchiolitis.” à Explain why breastfeeding is protective and the studies that support this thesis here and/or in the Discussions section. In addition to the one you mentioned, here is another important study:
o Nishimura, T., Suzue, J., & Kaji, H. (2009). Breastfeeding reduces the severity of respiratory syncytial virus infection among young infants: a multi-center prospective study. Pediatrics international : official journal of the Japan Pediatric Society, 51(6), 812–816. https://doi.org/10.1111/j.1442-200X.2009.02877.x
- “As treatments are limited, preventive measures are needed to reduce morbidity and mortality, especially from RSV infections.” à Increase the emphasis on preventive measures and mention Nirsevimab.
- “Standard oxygen supplementation in bronchiolitis is provided by nasal cannula (NC) and is upgraded to continuous positive airway pressure (CPAP) or invasive ventilation, if needed.” à Insert a reference.
- “In recent years, High-Flow Nasal Cannula oxygen therapy (HFNC), a non-invasive respiratory support supply, has emerged as a promising method to provide oxygen to children with bronchiolitis.” à Insert a reference.
- “Evidence suggests that the use of HFNC in bronchiolitis is limited to rescue therapy after failure of standard NC only in infants who are hypoxic.” à What evidence? Precise references must be included when these statements are made.
- “Different from previously, recent available data show no significant benefits for children treated with HFNC com-pared with NC or CPAP and suggest that HFNC produces no relevant differences in du-ration of hospitalization, days with oxygen supplementation and rate of admissions to Pediatric Intensive Care Unit () compared to NC.” à PICUs have already been mentioned; please, replace "Pediatric Intensive Care Unit ()" with "PICUs".
2. Materials and Methods
- Insert a subsection "2.1. Study design; patients enrollment".
- Write down where the study took place, including geographical references that anyone can understand and recognize.
- Why did you choose the months of September to March? Why didn't you select April, as well? Clarify this point by citing any other articles that have made the same choice.
- There is no mention of an Ethics Committee's approval at the end of section 2.1. This approval is absolutely required.
- “2.2. Statistical analysis”
- Indicate which cases the Fisher test was used in.
- Why did you use the ANOVA test if you had variables that were not normally distributed? Justify your decision. Depending on the situation, other tests, such as Kruskal-Wallis or Wilcoxon, are usually preferred.
- There is no description of the test used for multivariable analysis. Important distinction: multivariable, not multivariate.
3. Results
- “(table 4)” à It can't be 4, if it's quoted after table 1. Renumber, if necessary, all tables.
- “One patient aged < 1 month resulted positive for both Coronavirus SARS-CoV2 and RSV” à The term "Coronavirus" must be eliminated; it is already in the acronym "SARS-CoV2."
- In the current table 3, reformat the data for the total of the cohort, RSV+ and RSV-, inserting a row with the total, perhaps the second row. Also, better check the percentages in parentheses, there is an error in RSV- without fever.
- “Figure 1 and Figure 2 report the probability of treatment success (discharge to home without transfer to a Hub hospital with PICU or NICU): 96% for all patients (95% CI 93-99), 94% (95% CI 87-98) for RSV-positive and 98% (95% CI 95-100, P=NS) for RSV-negative patients when we stratify according to RSV infection.” à Figure 2 actually says 93% for RSV+. Check better, because the error is in the figures or text and in the current table 5.
- Table 4 à “RSV” instead of “VRS”.
- “On multivariate analysis, the patient's age (P=0.022) and the need of HFNC support (P<0.0000001) have been confirmed to be independent factors. (Table 6)” à "Multivariable" and not "multivariate". Furthermore, there is no mention of the fact that R2 is so low in almost all the variables in the system that it makes the model very questionable. Finally, Table 6 lacks an in-depth description of what has been put in the table; It doesn't seem like a logistical regression to me, what has been done? R2 must be described, together with SE and F, already described.
4. Discussion
- “(September-March 2021-2023).” à It is not clearly written, it seems that subjects hospitalized from September 2021 to March 2023 have been included.
- “Among risk factors, bronchodysplasia was detected in 0.5% and prematurity in 13% of infants. This rate was lower than those reported in recent Italian and USA studies (24.5% and 33%, respectively)” à There hasn't been much thought given to this section. The cited Italian study, for example, observes that the estimates precede the use of Pavilizumab. It would be appropriate to comment on your data, such as assuming that the preventive measures were effective given the 10% reduction in your study.
- “Among protective factors for infections, we observed that about half of RSV-negative patients were breastfed.” à I reiterate that this aspect requires more attention.
- “A recent Italian study of Istituto Superiore di Sanità demonstrated a change in RSV subtype circulation in the post-pandemic period.” àIn this regard, the "RespiVirNet" model of international and national virological and epidemiological surveillance should be mentioned. Furthermore, I invite you to read the following article, which discusses the importance of surveillance, the challenges posed by RSV, and the vaccination/preventive strategies mentioned in the conclusions section:
o Tramuto, F., Maida, C. M., Di Naro, D., Randazzo, G., Vitale, F., Restivo, V., Costantino, C., Amodio, E., Casuccio, A., Graziano, G., Immordino, P., & Mazzucco, W. (2021). Respiratory Syncytial Virus: New Challenges for Molecular Epidemiology Surveillance and Vaccination Strategy in Patients with ILI/SARI. Vaccines, 9(11), 1334. https://doi.org/10.3390/vaccines9111334
- “Despite a trend towards a higher incidence of RSV+ bronchiolitis in the total breastfed population, most RSV-positive infants less than <1 month […]” à “Younger than” instead of “less than”.
- “Multivariate analysis confirmed that age <1 month and the need of HFNC support were independently associated with a worse clinical outcome and a more frequent need for transfer to Hub hospital with a PICU or a NICU, according to literature.” à Again, "Multivariable". Also, I would be very cautious in saying "confirmed": as I said before, these are results to be read with caution, given the R2 almost everywhere very low.
- A "Limitations" section is missing. The study has several limitations, in terms of the size of the sample analyzed, and the analysis performed, just to give two examples, not to mention the fact that it is not possible to generalize the data since they come from a restricted geographical area with its specific characteristics and seasonality.
5. Conclusions
- “In our series, the majority of patients hospitalized for RSV bronchiolitis were patients not at risk. A different prophylactic strategy is necessary to reduce the social costs of this disease.” à A very strong statement, it needs to be argued further.
Comments on the Quality of English LanguageIn the previous section, I have already cited them.
Author Response
We thank all the reviewers because their helpful comments and suggestions have greatly improved the quality of our paper. We replied to all queries according to reviewer’s instructions.
Reviewer 1
Thank you for your review. We have improved our manuscript based on your suggestions.
JOURNAL TEMPLATE
About the wrong journal template, this was a mistake made by the editorial office that is now amended (attached please find the manuscript with the correct template).
TITLE
We corrected the title by contextualizing the study geographically and removing the capitalization: “Management of acute bronchiolitis in spoke hospitals in Northern Italy: analysis and outcome”.
ABSTRACT
We revised the abstract according to your suggestions.
KEYWORDS
We corrected the keywords and removed the running title. “Keywords: bronchiolitis; children; respiratory syncytial virus; high flow nasal cannula”.
INTRODUCTION
We have revised the introduction, organizing it according to your suggestions. We explained the protective mechanism of breastfeeding on bronchiolitis, and we increased the emphasis on preventive measures, mentioning Nirsevimab. References have been corrected and implemented. All corrections are underlined in the manuscript.
MATERIALS AND METHODS
- We included subsection 2.1 (“2.1 Study design; patient enrollment”) and better specified that the study took place in Piedmont, Northern Italy.
- About the selected study period, we chose the period between September and March because SARS-COV2 pandemic has changed the epidemiology of RSV, and an earlier season of RSV infection has been described starting from September 2021. The following are some studies that analyzed RSV bronchiolitis from September to March:
- Nenna R, Matera L, Licari A, Manti S, Di Bella G, Pierangeli A, Palamara AT, Nosetti L, Leonardi S, Marseglia GL, Midulla F; ICHRIS Group. An Italian Multicenter Study on the Epidemiology of Respiratory Syncytial Virus During SARS-CoV-2 Pandemic in Hospitalized Children. Front Pediatr. 2022 Jul 14;10:930281. doi: 10.3389/fped.2022.930281.
- Berdah, L.; Romain A.S.; Rivière, S.; Schnuriger, A.; Perrier, M.; Carbajal, R.; Lorrot, M.; Guedj, R.; Corvol, H. Retrospective observational study of the influence of the COVID-19 outbreak on infants' hospitalisation for acute bronchiolitis. BMJ Open 2022, 12, e059626. doi: 10.1136/bmjopen-2021-059626.
- Curatola, A.; Graglia, B.; Ferretti, S.; Covino, M.; Pansini, V.; Eftimiadi, G.; Chiaretti, A.;, Gatto, A. The acute bronchiolitis rebound in children after COVID-19 restrictions: a retrospective, observational analysis. Acta Biomed 2023, 13, 94, e2023031, doi: 10.23750/abm.v94i1.13552.
- Ghirardo S, Ullmann N, Ciofi Degli Atti ML, Raponi M, Cutrera R. Delayed season's onset and reduction of incidence of bronchiolitis during COVID-19 pandemic. Pediatr Pulmonol. 2021 Aug;56(8):2780-2781. doi: 10.1002/ppul.25461
- About the Ethics committee’s approval: we specified that “The study was conducted in full conformity with the principles of the Declaration of Helsinki. In accordance with current regulations, this research is not among those requiring a formal opinion from the ethics committee. This is a secondary use of data for research purposes for which specific informed consent was requested ab initio from patients who would be undertaking a treatment process.”
- “2.2. Statistical analysis”
- Indicate which cases the Fisher test was used in. Thank you for the questions. Fisher' exact test is a statistical significance test used in the analysis of contingency tables. Although in practice it is employed when sample sizes are small, it is valid for all sample sizes.
- Why did you use the ANOVA test if you had variables that were not normally distributed? Justify your decision. Depending on the situation, other tests, such as Kruskal-Wallis or Wilcoxon, are usually preferred. Thank you for this question. The Kruskal-Wallis one-way ANOVA is a non-parametric method for comparing k independent samples. It was a mistake in the last version of the paper. Differences were calculated with Kriskall-Wallis test.
- There is no description of the test used for multivariable analysis. Important distinction: multivariable, not multivariate. Thank you for the question. We added how logistic regression is used to predict the categorical dependent variable. It's used when the prediction is categorical, for example, yes or no, true or false, 0 or 1. This type of statistical model (also known as logit model) is often used for classification and predictive analytics. Logistic regression estimates the probability of an event occurring, such as voted or didn’t vote, based on a given dataset of independent variables. Since the outcome is a probability, the dependent variable is bounded between 0 and 1. In logistic regression, a logit transformation is applied on the odds—that is, the probability of success divided by the probability of failure. This is also commonly known as the log odds, or the natural logarithm of odds.
RESULTS
We corrected the results based on your suggestions. All corrections are underlined in the manuscript.
DISCUSSION
We modified the discussion according to your suggestions (corrections are underlined in the manuscript):
- We have specified the period in which the study took place: from September 1, 2021 to March 31, 2022 and from September 1, 2022 to March 31, 2023.
- We commented better on the low rate of prematurity observed in our study.
- We emphasized the role of breastfeeding.
- We described the epidemiological change of RSV after the pandemic and the importance of epidemiological surveillance and prevention.
- We introduced a section about limitations of the study
CONCLUSIONS
We modified the conclusions according to your suggestions (corrections are underlined in the manuscript).
Reviewer 2 Report
Comments and Suggestions for Authors
The authors hypothesized that the differences in HFNC use (more frequent in RSv+ patients) was related to disease severity. There is no mention of how HFNC was prescribed (was it only on physician judgment, or was it based on objective signs included in a guideline (used as a “treatment escalation”)? Could knowing RSV diagnosis induce the physician to use HFNC more “easily”?
If HFNC was indicated in more severe cases, authors cannot argue “being HFNC treatment an indicator of illness severity”. It could be an indicator of a poor outcome (need of transfer to 3er level hospital/PICU), but the illness severity decided the HFNC use.
This is not a good design to prove that HFNC use prevents the need to transfer to a 3er level hospital/PICU. There are several RCTs proving that HFNC does not modify the clinical outcome in infants with bronchiolitis.
Authors mentioned that “To provide uniformity of care with an evidence-based approach, in 2013 the first “Diagnostic Therapeutic Assistance Pathway for bronchiolitis in pediatric age” was drawn up. Afterwards, it has been updated in 2016 and 2020.”, but there is no link to that document. On the other hand, references included two documents that I believed are bronchiolitis guidelines in force when the study was carried out (references #11 and #14).
Patients belongs to two different (and consecutives) RSV seasons. Despite RSV is quite typical, since COVID-19 pandemic altered RSV seasonality and other issues, it would be interesting to have a table comparing patients of both seasons, in their main characteristics.
The Conclusions must be reviewed:
I am not sure that In this study, age less than 1 month and the need of treatment with HFNC were prognostic factors associated with a worse outcome, since HFNC was used in more severe patients.
Also, this study did not prove that “use of HFNC…. reduces transfer to a Hub hospital”. The study is not designed to assess that objective.
No “prophylactic strategy” was under study, to state that “another one is necessary to reduce the social costs of this disease”.
Table 1: Re-order (and re-name) variables sex/age/prematurity/weight-for-age
Fig 1 and Fig 2: Variable name is necessary on the X ax. On the Y ax, if “probability” is displayed numbers must be shown in decimals and not percentages.
Author Response
Reviewer 2
We thank all the reviewers because their helpful comments and suggestions have greatly improved the quality of our paper. We replied to all queries according to reviewer’s instructions.
- 1. The authors hypothesized that the differences in HFNC use (more frequent in RSV+ patients) was related to disease severity. There is no mention of how HFNC was prescribed (was it only on physician judgment, or was it based on objective signs included in a guideline (used as a “treatment escalation”)? Could knowing RSV diagnosis induce the physician to use HFNC more “easily”? In all patients, HFNC was used as treatment escalation, after failure of low-flow oxygen supplementation, regardless by knowing RSV diagnosis. Thank you for your question. We provided to better explain the concept. we modified the statement, writing " being HFNC treatment an indicator of poor outcome (need to transfer to a third level hospital/PICU" (and not "being HFNC treatment an indicator of illnes severity"). Corrections are underlined in the manuscript.
- Authors mentioned that “To provide uniformity of care with an evidence-based approach, in 2013 the first “Diagnostic Therapeutic Assistance Pathway for bronchiolitis in pediatric age” was drawn up. Afterwards, it has been updated in 2016 and 2020.”, but there is no link to that document. On the other hand, references included two documents that I believed are bronchiolitis guidelines in force when the study was carried out (references #11 and #14).Thank you for your suggestion. We provided to better explain the concept. Our PDTA (Percorso Diagnostico Terapeutico Assisitenziale) on bronchiolitis was developed in accordance with Hub hospital and is based on National guidelines on bronchiolitis so it was not attached. Italian guidelines were included in the references (Manti, S.; Staiano, A.; Orfeo, L.; Midulla, F.; Marseglia, G.L.; Ghizzi, C.; Zampogna, S.; Carnielli, V.P.; Favilli, S.; Ruggieri, M.; et al. UPDATE-2022 Italian guidelines on the management of bronchiolitis in infants. Ital J Pediatr 2023, 49, 19: 1-19: 18, doi: 10.1186/s13052-022-01392-6; Baraldi, E.; Lanari, M.; Manzoni, P.; Rossi, G.A.; Vandini, S.; Rimini, A.; Romagnoli, C.; Colonna, P.; Biondi, A.; Biban, P.; et al. Intersociety consensus document on treatment and prevention of bronchiolitis in newborns and infants. Ital J Pediatr 2014, 24, 65: 1-65: 13, doi: 10.1186/1824-7288-40-65).
- Patients belongs to two different (and consecutives) RSV seasons. Despite RSV is quite typical, since COVID-19 pandemic altered RSV seasonality and other issues, it would be interesting to have a table comparing patients of both seasons, in their main characteristics. This is a very important point. A comparative table of patient characteristics in the two seasons analyzed was not done previously because both seasons referred to the post-pandemic period. We are writing another paper focusing on this seasonality between the pre- and post-pandemic period.
- Table 1: Re-order (and re-name) variables sex/age/prematurity/weight-for-ageTable 1: According to your suggestion, we re-named sex/age/prematurity/weight-for-age variables in tables 1 and 5.
.
Reviewer 3 Report
Comments and Suggestions for Authors
Descriptive study with no innovative or novel findings, but likely relevant for the local or regional audience or policy makers or from an epidemiological viewpoint.
What was the number given to the study by the ethical committee? How can the parents have provided a specific consent form in this retrospective study?
I would invite the authors to provide relevant definitions for the different variables or outcome parameters in the methods section. For instance: 'treatment success, complication, prematurity,...' One can find these in several parts of the manuscript, but it is better to put them together here so that the reader has a clear idea on what they exactly mean.
I did not find any data regarding passive RSV vaccination (Palivizumab). Are these data known for this cohort or for the region as a whole?
In the discussion there is suddenly a paragraph on breastfeeding, yet I did not find the data for this cohort in the M&M section or the result section (not in tables).
Comments on the Quality of English Language
Some minor English editing is needed.
e.g. data are not 'resumed' in a table but 'summarized'.
Author Response
Responses to reviewer 3
We thank all the reviewers because their helpful comments and suggestions have greatly improved the quality of our paper. We replied to all queries according to reviewer’s instructions.
Descriptive study with no innovative or novel findings, but likely relevant for the local or regional audience or policy makers or from an epidemiological viewpoint. Yes, we agree that we report no innovative findings, but for many families the local treatment is a great help.
What was the number given to the study by the ethical committee? Thank you for this important question. The study was conducted in full conformity with the principles of the Declaration of Helsinki. In accordance with current regulations, this research is not among those requiring a formal opinion from the ethics committee. This is a secondary use of data for research purposes for which specific informed consent was requested ab initio from patients who would be undertaking a treatment process.
How can the parents have provided a specific consent form in this retrospective study? The parents or legal guardians of all patients had signed consent at the time of admission. This consent provides for the management and processing of personal data, of course made anonymous. This consent is obviously optional and is the expression of the ASLTO4 desire to encourage the collection of clinical and laboratory data.
I would invite the authors to provide relevant definitions for the different variables or outcome parameters in the methods section. For instance: 'treatment success, complication, prematurity,…' One can find these in several parts of the manuscript, but it is better to put them together here so that the reader has a clear idea on what they exactly mean. Thank you for these questions:
- treatment success was defined as possibility to discharge to home (integrated into the text)
-complication were defined as events outside the normal course of the disease (integrated into the text)
-prematurity was defined as any child who were born before the 37+0 weeks of pregnancy (integrated into the text).
I did not find any data regarding passive RSV vaccination (Palivizumab). Are these data known for this cohort or for the region as a whole? Thank you for this question. Unfortunately, passive vaccination data with Palivizumab in this patient cohort are not known.
In the discussion there is suddenly a paragraph on breastfeeding, yet I did not find the data for this cohort in the M&M section or the result section (not in tables). Yes, thank you. The data on breastfeeding are illustrated in the paragraph below Table 1. These data were subsequently discussed in the relevant part, as the protective role of breastfeeding against viral infections is known from the literature.
Round 2
Reviewer 1 Report
Comments and Suggestions for Authors
I am glad to read that the suggestions and comments raised were recognized and implemented; I believe the manuscript has improved and is now worth publishing.
Reviewer 2 Report
Comments and Suggestions for Authors
no comments